# Biosynthesis and Characterization of Gold Nanoparticles Produced Using *Rhodococcus* Actinobacteria at Elevated Chloroauric Acid Concentrations

**DOI:** 10.3390/ijms232112939

**Published:** 2022-10-26

**Authors:** Maria S. Kuyukina, Marina V. Makarova, Irena B. Ivshina, Konstantin P. Kazymov, Boris M. Osovetsky

**Affiliations:** 1Microbiology and Immunology Department, Perm State University, 614990 Perm, Russia; 2Institute of Ecology and Genetics of Microorganisms, Perm Federal Research Center, Ural Branch of the Russian Academy of Sciences, 614081 Perm, Russia; 3Mineralogy and Petrography Department, Perm State University, 614990 Perm, Russia

**Keywords:** actinobacteria, *Rhodococcus*, chloroauric acid, gold nanoparticles, biosynthesis, zeta potential, antimicrobial activity

## Abstract

The growing industrial and medical use of gold nanoparticles (AuNPs) requires environmentally friendly methods for their production using microbial biosynthesis. The ability of actinobacteria of the genus *Rhodococcus* to synthesize AuNPs in the presence of chloroauric acid (HAuCl_4_) was studied. The effect of elevated (0.8–3.2 mM) concentrations of HAuCl_4_ on bacterial viability, morphology, and intracellular accumulation of AuNPs by different *Rhodococcus* species was shown. An increase in surface roughness, a shift of the zeta potential to the positive region, and the formation of cell aggregates of *R. erythropolis* IEGM 766 and *R. ruber* IEGM 1135 during nanoparticle synthesis were revealed as bacterial adaptations to toxic effects of HAuCl_4_. The possibility to biosynthesize AuNPs at a five times higher concentration of chloroauric acid compared to chemical synthesis, for example, using the citrate method, suggests greater efficiency of the biological process using *Rhodococcus* species. The main parameters of biosynthesized AuNPs (size, shape, surface roughness, and surface charge) were characterized using atomic force microscopy, dynamic and electrophoretic light scattering, and also scanning electron microscopy in combination with energy-dispersive spectrometry. Synthesized by *R. erythropolis* spherical AuNPs have smaller (30–120 nm) dimensions and are positively (12 mV) charged, unlike AuNPs isolated from *R. ruber* cells (40–200 nm and −22 mV, respectively). Such differences in AuNPs size and surface charge are due to different biomolecules, which originated from *Rhodococcus* cells and served as capping agents for nanoparticles. Biosynthesized AuNPs showed antimicrobial activity against Gram-positive (*Micrococcus luteus*) and Gram-negative (*Escherichia coli*) bacteria. Due to the positive charge and high dispersion, the synthesized by *R*. *erythropolis* AuNPs are promising for biomedicine, whereas the AuNPs formed by *R*. *ruber* IEGM 1135 are prone to aggregation and can be used for biotechnological enrichment of gold-bearing ores.

## 1. Introduction

In the last two decades, technologies for producing gold nanoparticles (AuNPs) with desired properties for various industries and medicine have rapidly advanced [1]. The growing biomedical use of AuNPs as antibacterial agents, modulators of enzymatic activity, and antitumor agents encourages the search for environmentally safe methods of their biological synthesis without toxic intermediates related to chemical synthesis [2,3].

Biosynthesis of nanometals using microorganisms and plants has a number of advantages over widespread chemical and physical methods, in particular, savings of reducing agents and energy, and high process performance [1,3]. A large-scale production of colloidal gold by chemical methods usually requires elevated temperatures or harsh chemicals as well as energy and capital-intensive synthesis procedures to produce monodisperse nanoparticles [4]. Biologically synthesized AuNPs exhibit enhanced stability and better control of morphology; they are already functionalized with biomolecules, performing biocompatibility and important physiological functions, such as antimicrobial and antibiofilm activities [1,2]. Biogenic nanogold formed as a result of the vital activity of bacteria and the subsequent aggregation of nanoparticles, presents part of the reserves in the world’s largest deposits (Witwatersrand, Karlin) and, in a certain amount, is extracted by modern technologies [5].

There are literature data on the ability of bacteria, fungi, and microalgae to recover gold from gold-containing compounds, namely chloroauric acid (HAuCl_4_), with the formation of nanoparticles [6,7,8,9,10]. However, the spectrum of the studied microorganisms is limited; physicochemical and antimicrobial properties of biosynthesized AuNPs are insufficiently studied. To our knowledge, the reported biosynthesis using microorganisms occurs at a concentration of HAuCl_4_ up to 1 mM due to its toxicity. Therefore, resistance to toxic effects of HAuCl_4_ is a necessary prerequisite for a nanogold-producing strain. 

Actinobacteria of the genus *Rhodococcus* are highly resistant to adverse environmental factors, including the toxic eco-pollutant’s exposure, capable of accumulating heavy metal ions and synthesizing non-toxic biosurfactants with metal-chelating properties [11,12,13]. However, the ability of these microorganisms to produce nanometals via bioconversion of metal salts is just beginning to be explored [12]. There is a single published report [14] on the interaction of a non-identified *Rhodococcus* strain with chloroauric acid, which led to the reduction of [AuCl_4_]^−^ ions and the formation of monodisperse AuNPs. Thus, selection of appropriate *Rhodococcus* species resistant to high concentrations of HAuCl_4_ in order to increase nanoparticle biosynthesis and thorough characterization of the synthesized AuNPs are essential for the development of industrial nanogold bioproduction. 

A plethora of analytical methods are used to determine morphology and physicochemical characteristics of nanoparticles (bio)synthesized, among them electron (scanning and transmission) microscopy coupled with spectrometry is the most powerful but expensive. Therefore, a comparative study using less expensive methods, such as atomic force microscopy and light scattering, would help expand the range of relevant and available methods for nanoparticle characterization. 

The aim of the work is to study the ability of different *Rhodococcus* species to synthesize AuNPs in the presence of elevated (0.8–3.2 mM) concentrations of chloroauric acid as well as to analyze the obtained nanoparticles for their physicochemical properties (size, shape, surface roughness, surface charge) and antimicrobial activity.

## 2. Results

### 2.1. Resistance of Rhodococcus Strains to Chloroauric Acid and Biosynthesis of AuNPs

The results of cell viability testing using staining with colorless iodonitrotetrazolium (INT), reduced by dehydrogenase enzymes of the electron transfer system into red-violet formazan, which was quantified spectrophomerically by optical density (OD) at 630 nm, are shown in Figure 1. According to data in Figure 1, at low (0.1–0.4 mM) concentrations of chloroauric acid, the viability of *Rhodococcus* cells was comparable with that of the biotic control, while an increase in the acid concentration up to 1.6 mM led to a sharp increase in OD_630_ of the INT-stained cell suspension, indicating a nearly 2-fold increase in the respiratory activity of bacterial cells. At the maximum (3.2 mM) concentration of HAuCl_4_, only a slight (5%) decrease in cell viability was observed. Thus, the tested *Rhodococcus* strains showed higher resistance to chloroauric acid compared to other microorganisms [5,7]. For example, at a HAuCl_4_ concentration above 1 mM, *Anabaena* sp. did not form AuNPs due to the death of cyanobacteria and a disrupted functioning of the enzyme system [7]. 

In the presence of chloroauric acid (≥0.8 mM), cells of *R. erythropolis* IEGM 766 aggregated to form large (100 to 200 µm) agglomerates, while *R. ruber* IEGM 1135 was characterized by the formation of smaller aggregates and the presence of individual cells (Figure 2).

Atomic force microscopy (AFM) scanning revealed changes in the surface relief of *Rhodococcus* cells incubated with chloroauric acid. Figure 3 shows that control cells of *R. ruber* IEGM 1135 had a relatively uniform and smooth surface (the RMS roughness was 133 nm). In the presence of HAuCl_4_, an increase in surface roughness to an average of 149 nm was observed. A similar increase in the cellular surface relief was observed for all the studied *Rhodococcus* species exposed to chloroauric acid. Other studies [15,16] also showed an increased roughness of the cell wall as an adaptation feature of *Rhodococcus* spp. under the influence of toxic dehydroabietic acid and sodium diclofenac.

As seen from Figure 4, zeta potential values of *Rhodococcus* cells exposed to chloroauric acid shifted from the negative to the positive region. The most pronounced changes in zeta potential (from −24 to 14.7 mV) were detected for *R. ruber* IEGM 1135 after 1 h of exposure to 1.6 mM HAuCl_4_. Interestingly, after 24 h, zeta potentials of *R. erythropolis* IEGM 766 and *R. ruber* IEGM 1135 cells remained positive, while *R. fascians* IEGM 525 cells become negatively charged. The revealed different dynamics of the cellular charge in the process of AuNPs biosynthesis might be determined by the individual characteristics of *Rhodococcus* strains and requires further study.

The dynamics of AuNPs biosynthesis by different *Rhodococcus* strains at increasing (0.2–3.2 mM) concentrations of chloroauric acid was examined. Figure 5 shows that the maximum synthesis of nanoparticles by *R. erythropolis* IEGM 766, *R. fascians* IEGM 525, and *R. ruber* IEGM 1135 was observed at a HAuCl_4_ concentration of 1.6 mM, whereas *R. rhodochrous* IEGM 1162 actively synthesized AuNPs in the presence of as little as 0.8 mM HAuCl_4_. To further characterize the biosynthesized AuNPs, *R. erythropolis* IEGM 766 and *R. ruber* IEGM 1135 strains were selected, which we featured by the highest survival rate and different mechanisms of resistance to elevated concentrations of HAuCl_4_.

### 2.2. Physicochemical Properties of Biosynthesized AuNPs

AFM analysis revealed (Figure 6) spherical nanoparticles of 60 ± 20 nm in size with smooth surfaces, synthesized by *R. erythropolis* IEGM 766, which tend to form micrometer-sized AuNPs aggregates. It should be noted that the spherical shape, which provides the minimum specific surface area, is characteristic of most gold nanoparticles synthesized by microorganisms [3].

The average hydrodynamic diameter of AuNPs determined by the method of dynamic light scattering (Figure 7) was smaller (~80 nm) for nanoparticles synthesized by *R. erythropolis* IEGM 766, but it was one and a half times larger for AuNPs obtained from *R. ruber* IEGM 1135 cells. According to the literature data [3,6,7,8,9,10], the sizes of biosynthesized AuNPs vary in a wide range (5 to 400 nm). The charge of biosynthesized AuNPs measured by electrophoretic light scattering also differed depending on the bacterial strain used. Thus, zeta potential of AuNPs synthesized by *R. erythropolis* IEGM 766 was positive (12 mV), whereas nanoparticles from *R. ruber* IEGM 1135 had a negative charge (−20 mV) higher in modulus, indicating their greater colloidal stability. Other studies also reported different zeta potential values of AuNPs [6,8,10], depending on the microorganism used. For example, zeta potential of AuNPs obtained using extracellular filtrate of the fungus *Pleurotus ostreatus* was −24 mV [8]. The surface charge of colloidal gold recovered in the presence of bacteria *Azospirillum brasilense* ranged from −5.08 to −3.34 mV [6]. AuNPs synthesized by an extract of the fungus *Lignosus rhinocerotis* and stabilized with chitosan had a positive surface charge ranging from 14.78 to 41.39 [10]. The authors explained the positively charged surface by the presence of chitosan molecules on the surface of AuNPs. It seems that the surface charge of biosynthesized AuNPs is determined by stabilizing and functionalizing biomolecules (proteins, polysaccharides or organic acids), which are diverse in different microorganisms [1,3].

Scanning electron microscopy (SEM) analysis of biosynthesized nanoparticles confirmed the presence of Au by a bright material contrast (Table 1). For reliable diagnosis of AuNPs, the chemical composition of nanoparticles was determined using energy-dispersive spectrometry. The presence of other elements (in addition to Au) in the obtained spectra can be explained by the large size of the electron probe (about 1 mm); as a result, the identified elements also included components (namely Si and O) of the glass slide with the sample of nanoparticles applied. In addition, the presence of C and O indicated organic compounds which originated from *Rhodococcus* cells and served as capping agents for nanoparticles. Although the composition of coating material was not determined, it was previously shown [17] that when exposed to metal stress, *Rhodococcus* biofilms produced an extracellular matrix with a consistently high lipid content and increased concentrations of polysaccharides and proteins, which may contribute to stabilization of nanoparticles.

Using SEM, the morphological features of AuNPs synthesized by different *Rhodococcus* strains were revealed (Figure 8). In particular, AuNPs from *R. erythropolis* IEGM 766 cells were represented by individual nanoparticles and had smaller (30–120 nm) sizes compared to AuNPs synthesized by *R. ruber* IEGM 1135 (40–200 nm), which tended to aggregate (Figure 9). Using transmission electron microscopy, Ahmad et al. [14] also showed that *Rhodococcus* sp. cells formed monodisperse AuNPs of 5–16 nm in size.

### 2.3. Antimicrobial Activity of Biosynthesized AuNPs

As shown in Figure 10, the obtained AuNPs had significant antimicrobial effects on the bacterial cultures tested. In particular, a 65% decrease in cell viability of *E. coli* ATCC 25922 and a less pronounced (by 51 and 18%) suppression of *M. luteus* IEGM 401 were revealed upon exposure to AuNPs synthesized by *R. erythropolis* IEGM 766 and *R. ruber* IEGM 1135, respectively. Another research group [9] also showed the inhibiting effect of AuNPs obtained from *Deinococcus radiodurans* on Gram-negative rather than Gram-positive bacteria, which, apparently, is due to differences in the cell wall structure.

## 3. Discussion

We investigated the ability of *Rhodococcus* species to intracellular synthesis of gold nanoparticles in the presence of high (up to 3.2 mM) concentrations of toxic chloroauric acid. Importantly, the possibility of AuNPs biosynthesis at a 5 times higher concentration of chloroauric acid compared to chemical synthesis, for example, using the citrate method [4], suggests greater efficiency of the biological process using actinobacteria of the genus *Rhodococcus*. In addition, mild reaction conditions (no elevated temperatures or aggressive chemicals are required) and easy recovery of synthesized nanoparticles without the use of organic solvents increase technological advantages of biosynthesis. Further research can be aimed at controlling the capping biomolecules, which determine largely the functions of AuNPs. Such control of cellular polysaccharide and protein synthesis would involve the selection of an appropriate *Rhodococcus* strain and modification of culture conditions, including carbon substrate and metal stress [15,16,17]. 

A plethora of analytical methods are used to determine characteristics (the most important are morphology and electric charge) of nanoparticles (bio)synthesized. Table 2 summarizes the data on the shape and size of biosynthesized AuNPs obtained in the present study using different methods (AFM, dynamic light scattering, and SEM). Apparently, SEM provided the most precise information related to the morphology of nanoparticles, especially in combination with elemental spectroscopic analysis [3,4]. Even so, because of the expensive equipment and the complexity of sample preparation, other methods are increasingly used to characterize nanoparticles. For example, AFM allows scanning aggregated nanoparticles or deposited on curved or rough substrates, as well as measuring the surface relief of nanoparticles and their adhesion to the cantilever probe, which provides additional information on nanomechanical properties of AuNPs [18]. In turn, the method of dynamic light scattering does not require sample preparation, has high data similarity with SEM (Table 2), and allows determining the size of nanoparticles suspended in liquid media without their deposition and drying. In addition, dynamic and electrophoretic light scattering methods implemented in one device (ZetaSizer Nano ZS, Malvern Instruments, UK) make it possible to determine the hydrodynamic diameter and zeta potential of AuNPs in one sample.

The revealed variations in the morphology and surface characteristics of nanoparticles, obtained using different *Rhodococcus* species, determine their possible applications. Thus, given their relatively small size, positive charge and recovery as individual nanoparticles, AuNPs synthesized by *R. erythropolis* IEGM 766 seem promising for biotechnology and medicine as antimicrobial agents. It should be noted that positively charged metal nanoparticles usually exhibit higher antimicrobial activity due to strong electrostatic interaction with the negatively charged microbial cell wall [19]. Biosynthesis of AuNPs prone to aggregation by R. ruber IEGM 1135 cells can be useful for the enrichment of gold-bearing ores and concentrates (at the stages of recovery and deposition) [20], as biotechnological methods of extracting fossil gold are becoming increasingly important. 

## 4. Materials and Methods

### 4.1. Microorganisms and Culture Conditions

Four *Rhodococcus* strains, namely *R. erythropolis* IEGM 766, *R. fascians* IEGM 525, *R. rhodochrous* IEGM 1162, and *R. ruber* IEGM 1135 from the Regional Specialized Collection of Alkanotrophic Microorganisms of the Institute of Ecology and Genetics of Microorganisms, Perm, Russia (IEGM; www.iegmcol.ru; WFCC/WDCM 768; UNU/CKP 73559/480868) were used for AuNPs biosynthesis. Bacteria grown in a Luria–Bertani (LB) medium at 28 °C for 48 h were washed with distilled water and suspended in water to an optical density (OD_600_) of 1.5–2.2.

To determine the antimicrobial properties of biosynthesized AuNPs, *Escherichia coli* ATCC 25922 and *Micrococcus luteus* IEGM 401 were used, which were grown in LB for 48 h at 37 and 28 °C, respectively.

### 4.2. Determination of Bacterial Viability and Ability to Reduce Chloroauric Acid into AuNPs

Aqueous solutions (0.1 to 3.2 mM) of chloroauric acid (Sigma−Aldrich, St. Louis, MO, USA) were prepared in 2-fold serial dilutions, added to 96-well microplates, which were inoculated with bacterial suspensions, and continuously stirred (300 rpm) on a microplate shaker (Heidolph, Schwabach, Germany) for 48 h at room temperature. Then, 0.2% iodonitrotetrazolium (INT) chloride (Sigma, St. Louis, USA) was added into microplates and left for 2 h to allow the INT reduction to insoluble red-violet INT-formazan in the presence of actively respiring cells. The concentration of formazan was measured spectrophotometrically using a microplate reader (Multiskan Ascent, Thermo, Vantaa, Finland) at 630 nm (OD_630_) to assess cell viability. The suitability of the INT-staining method for cell viability testing was confirmed for one strain, *R. ruber* IEGM 1135, by direct Nutrient Agar plating and colony forming unit (CFU) counting from experimental cell suspensions with different (0.2–3.2 mM) concentrations of chloroauric acid. Alternatively, cell suspensions incubated with chloroauric acid and unstained with INT were used to estimate the amount of biosynthesized AuNPs, since the appearance of a bright purple color measured by OD_630_ indicated the formation of gold nanoparticles due to the excitation of surface plasmon oscillations [21]. All experiments were made in 8-fold replications (eight microplate wells of each variant).

### 4.3. Biosynthesis and Recovery of AuNPs

For the AuNPs production, *Rhodococcus* cells were grown in 250 mL Erlenmeyer flasks containing 100 mL of LB on an orbital shaker (160 rpm) at 28 °C for 72 h. Cells were separated by centrifugation (3000× *g*) and washed twice with sterile distilled water. The collected biomass was suspended in 50 mL of water (OD_600_ 1.5–2.2) with the addition of 1.6 mM chloroauric acid in 250 mL Erlenmeyer flasks. The mixture was placed on a shaker (160 rpm), and the reaction was carried out at room temperature for 24 h. Three parallel flasks were processed for each *Rhodococcus* strain.

To recover AuNPs, the biomass was separated from the reaction mixture by centrifugation at 4000× *g* for 10 min and the precipitate was washed twice with deionized water. Given that the bright purple color characteristic of AuNPs was present only in the cell precipitate and was absent in the supernatant, a conclusion was drawn about the intracellular localization of synthesized AuNPs. This conclusion was confirmed by AFM scanning of rhodococcal cells immediate after the reaction with HAuCl_4_. In all cases, when four *Rhodococcus* strains were scanned, extracellular AuNPs were not detected. Therefore, in order to simplify the procedure, AuNPs were recovered directly from the reaction mixture without washing the biomass. Once the bacterial cells were disrupted using a Soniprep 150 ultrasonic disintegrator (MSE, London, UK) and centrifuged (MiniSpin, Eppendorf, Hamburg, Germany) at 4000× *g* for 10 min, the resulting supernatant was passed through a membrane filter (0.2 µm) and AuNPs were precipitated from the filtrate using a MiniSpin microcentrifuge at 6700 g for 1 h. The resulting AuNPs were washed twice with sterile deionized water prior to analysis.

### 4.4. Microscopic Examinations of Rhodococcus Cells and Biosynthesized AuNPs

The morphology of *Rhodococcus* cells exposed to chloroauric acid was examined using an Axiostar plus light microscope (Carl Zeiss, Jena, Germany) with phase contrast and an oil-immersion lens (100×). Five fields of view were examined for each experimental variant. A more detailed study of morphological parameters of bacterial cells and biosynthesized AuNPs was carried out using an MFP-3D-BIO atomic force microscope (AFM) (Asylum Research, Santa Barbara, CA, USA). Approximately 10 μL of a bacterial or AuNPs suspension in deionized water was deposited on a cover glass and allowed to dry. Images were acquired using Olympus AC240TS silicon cantilevers (Olympus, Tokyo, Japan) with resonance frequencies of 50–90 kHz and spring constants of 0.5–4.4 N/m and processed using the Igor Pro 6.22A (WaveMetrics, Lake Oswego, OR, USA) software. The dimensions (length and width) and root mean square (RMS) roughness of bacterial cells and AuNPs were calculated from the height images (minimum 20 bacterial cells and 20 nanoparticles of each variant). 

The shape and size of biosynthesized AuNPs were also analyzed by scanning electron microscopy (SEM) using a Quattro C (Thermo Fisher Scientific, Czech Republic) microscope with field emission. The elemental composition of AuNPs was analyzed by an Ultra Dry energy dispersion spectrometer from the same manufacturer. The following operating parameters were set: a distance of 10 mm, a voltage of 25 kV, and a current of 1.3 nA. Approximately 10 μL of AuNPs suspension in deionized water was deposited on a cover glass and allowed to dry. A thin conductive layer of carbon was sprayed onto the sample to remove the charge and shield the incident beam from the charge accumulated in the bulk of the material. Each sample was scanned at least three times in different parts. 

### 4.5. Hydrodynamic Diameter and Zeta Potential Measurements

The hydrodynamic diameter of AuNPs was measured by dynamic light scattering at an angle of 173° using a ZetaSizer Nano ZS (Malvern Instruments, Malvern, UK) analyzer. The electrokinetic (zeta) potential of *Rhodococcus* cells and biosynthesized AuNPs was measured by electrophoretic light scattering in 10 mM KNO_3_ (pH 5.5–6). All experiments were performed in three replications.

### 4.6. Antimicrobial Activity of Biosynthesized AuNPs

Antimicrobial activity of the biosynthesized AuNPs was tested against Gram-negative and Gram-positive bacteria. The viability of *E. coli* ATCC 25922 and *M. luteus* IEGM 401 cells influenced by AuNPs was determined by INT staining, as described above. The percentage of viable cells was calculated from the OD_630_ difference between the experimental and control (without nanoparticles) bacterial suspensions. The abiotic control (AuNPs without bacteria) was also performed and resulted in very low OD_630_ of 0.06–0.07.

## Figures and Tables

**Figure 1 ijms-23-12939-f001:**
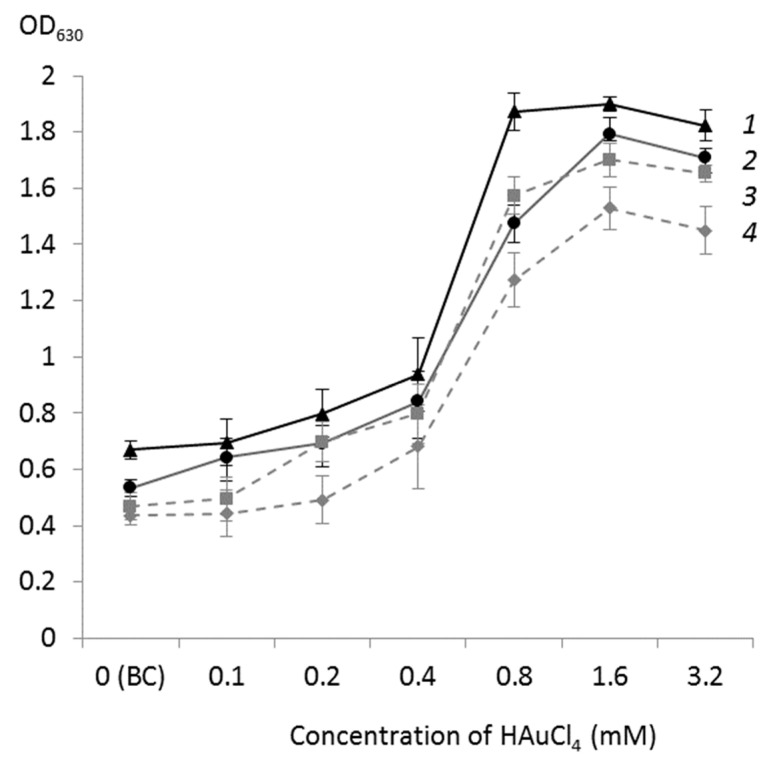
Effects of different HAuCl_4_ concentrations on the viability of *Rhodococcus* spp. (OD_630_ values of cell suspensions after staining with INT). BC—biotic control, (1)—*R. erythropolis* IEGM 766, (2)—*R. ruber* IEGM 1135, (3)—*R. rhodochrous* IEGM 1162, and (4)—*R. fascians* IEGM 525.

**Figure 2 ijms-23-12939-f002:**
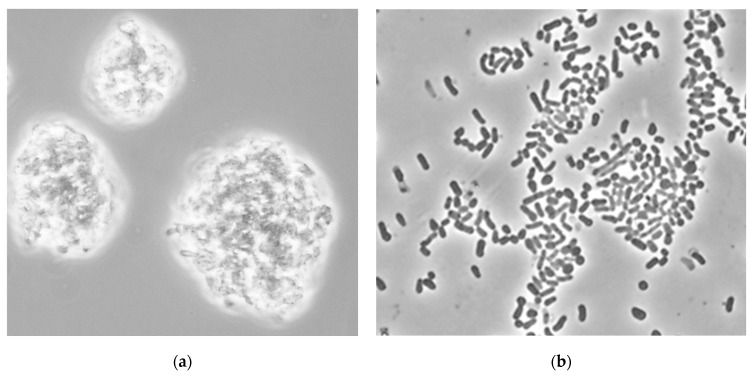
Phase-contrast microscopy of cells of *R. erythropolis* IEGM 766 (**a**) and *R. ruber* IEGM 1135 (**b**) exposed to HAuCl_4_ (magnification ×1000).

**Figure 3 ijms-23-12939-f003:**
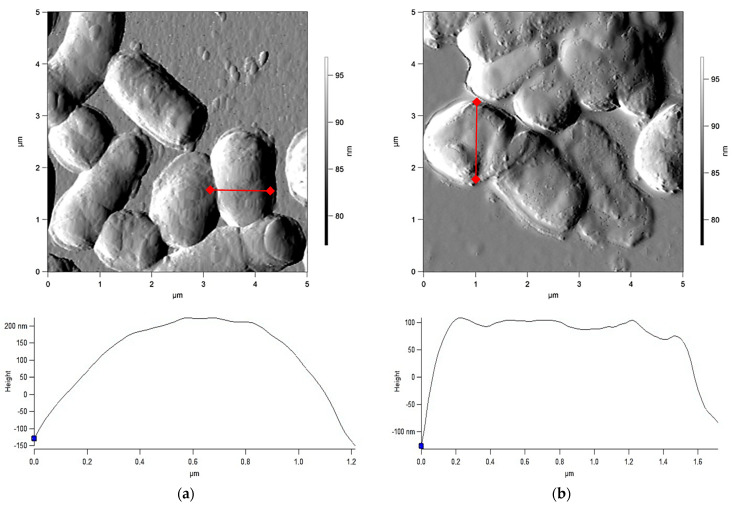
AFM images and profilometry of *R. ruber* IEGM 1135. Control cells (**a**) and cells exposed to HAuCl_4_ (**b**).

**Figure 4 ijms-23-12939-f004:**
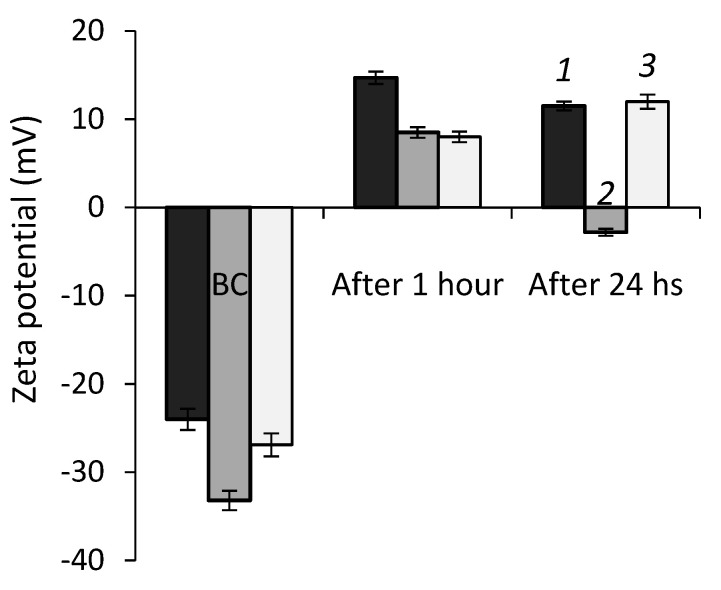
Zeta potential values of *Rhodococcus* cells after exposure to HAuCl_4_ for 1 h and 24 h. BC—biotic control. (1)—*R. erythropolis* IEGM 766, (2)—*R. ruber* IEGM 1135, and (3)—*R. fascians* IEGM 525.

**Figure 5 ijms-23-12939-f005:**
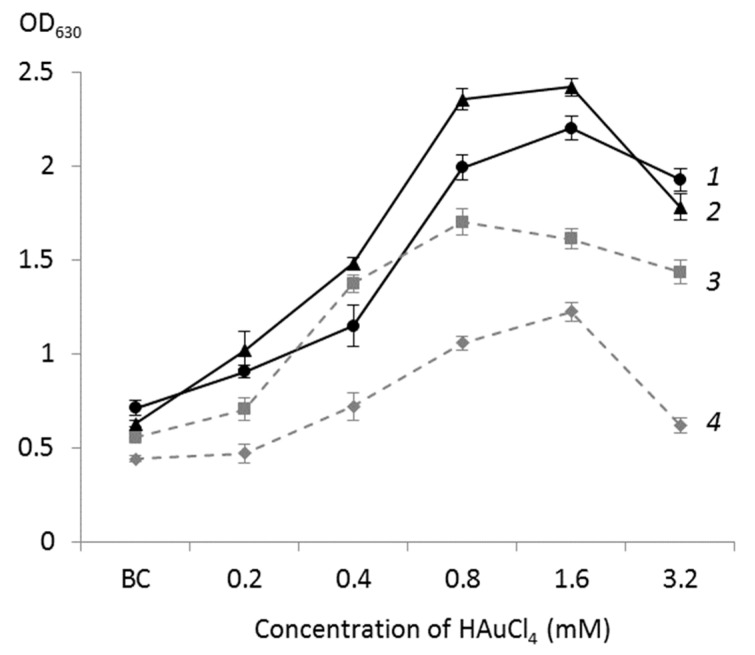
Biosynthesis of AuNPs (OD_630_ values) by *Rhodococcus* spp. in the presence of different HAuCl_4_ concentrations. BC—biotic control, (1)—*R. ruber* IEGM 1135, (2)—*R. erythropolis* IEGM 766, (3)—*R. rhodochrous* IEGM 1162, and (4)—*R. fascians* IEGM 525.

**Figure 6 ijms-23-12939-f006:**
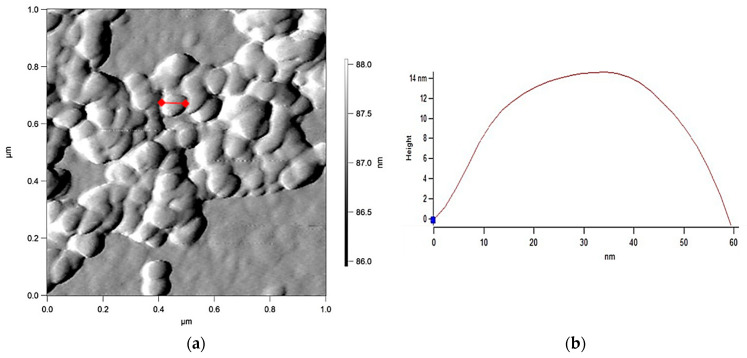
AFM images (**a**) and profilometry (**b**) of AuNPs synthesized by *R. erythropolis* IEGM 766.

**Figure 7 ijms-23-12939-f007:**
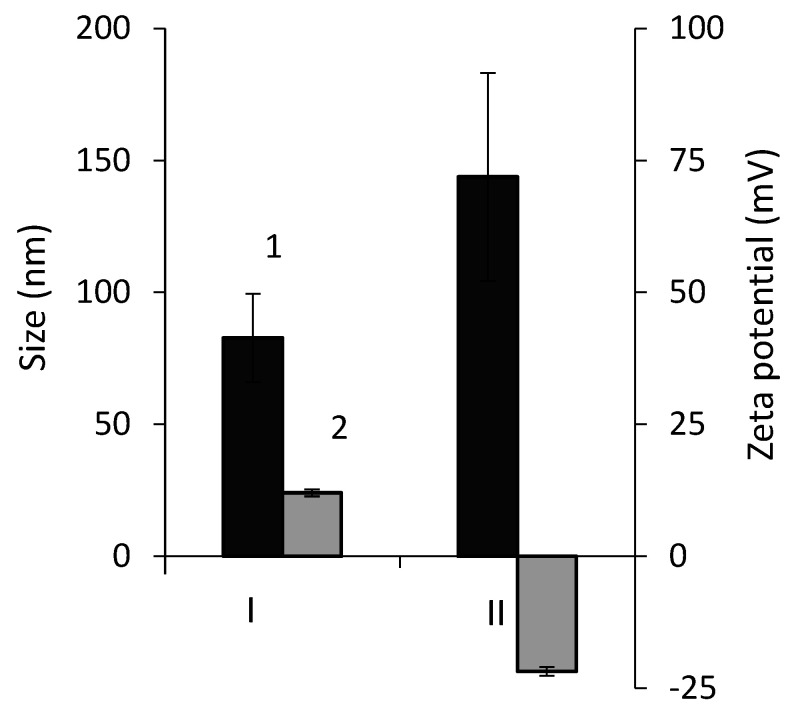
Hydrodynamic diameter (1) and zeta potential (2) values of AuNPs synthesized by *R. erythropolis* IEGM 766 (I) and *R. ruber* IEGM 1135 (II).

**Figure 8 ijms-23-12939-f008:**
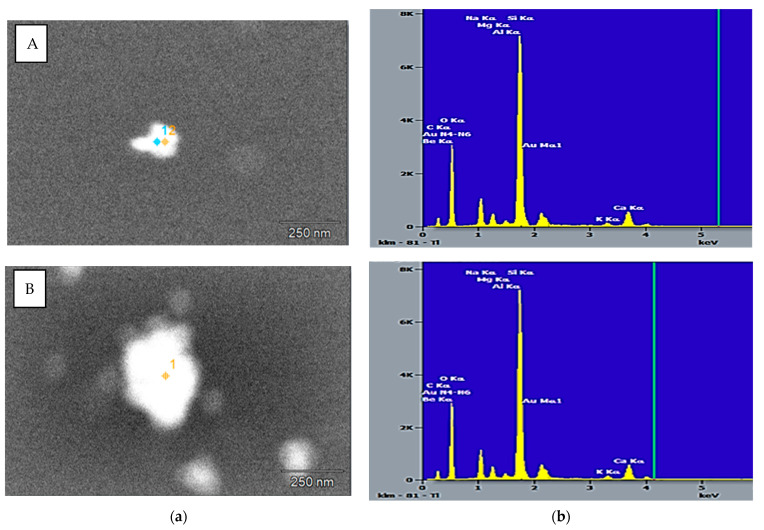
SEM images (**a**) and elemental spectra (**b**) of AuNPs synthesized by *R. erythropolis* IEGM 766 (**A**) and *R. ruber* IEGM 1135 (**B**). 1, 2—electron probe markers.

**Figure 9 ijms-23-12939-f009:**
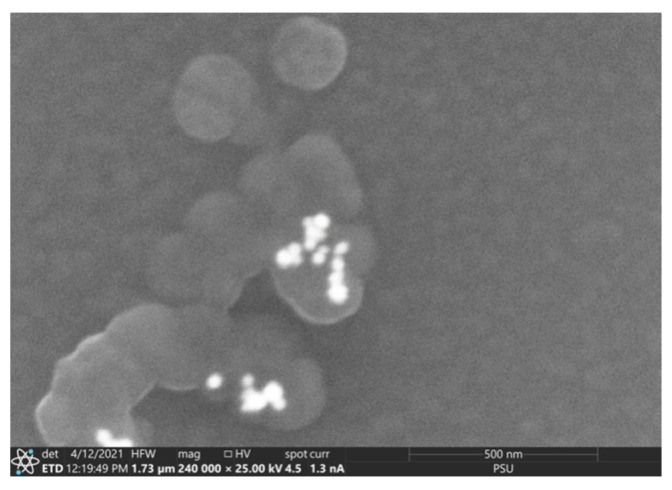
The SEM image shows aggregates of AuNPs (white color) synthesized by *R. ruber* IEGM 1135.

**Figure 10 ijms-23-12939-f010:**
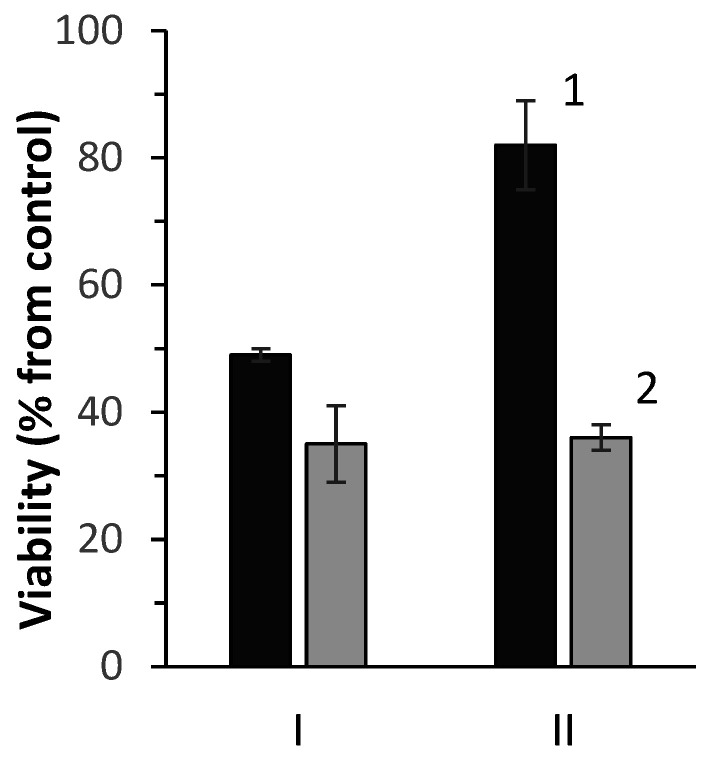
Antimicrobial activity of AuNPs synthesized by *R. erythropolis* IEGM 766 (I) and *R. ruber* IEGM 1135 (II) against *M. luteus* IEGM 401 (1) and *E. coli* ATCC 25922 (2).

**Table 1 ijms-23-12939-t001:** Elemental composition (%) of the SEM preparations with biosynthesized AuNPs.

Rhodococcus Strains	Be	C	O	Na	Mg	Al	Si	K	Ca	Au
R. erythropolis IEGM 766	1.76	6.89	34.38	6.14	2.02	0.51	29.57	0.69	4.69	13.35
R. ruber IEGM 1135	2.4	5.13	36.65	5.84	1.86	0.52	29.48	0.82	4.74	12.56

**Table 2 ijms-23-12939-t002:** Comparative summary of shape and size (nm) of biosynthesized AuNPs as measured by different methods.

Parameter/Method	AFM	Dynamic Light Scattering	SEM
AuNPs synthesized by *R. erythropolis* IEGM 766			
Shape	Spherical	-	Spherical
Size	50–90	40–120	30–120
AuNPs synthesized by *R. ruber* IEGM 1135			
Shape	Spherical	-	Spherical
Size	60–180	120–160	40–200

## Data Availability

Not applicable.

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
