# Peer review of "Biosynthesis and Characterization of Gold Nanoparticles Produced Using Rhodococcus Actinobacteria at Elevated Chloroauric Acid Concentrations"

_ijms, 2022, doi:10.3390/ijms232112939_

Round 1

Reviewer 1 Report

Dear Authors,

The paper is interesting and show a different approach of measuring the AuNPs through chance in optical density. The idea is intriguing and well conducted. However, there is few thing that bother me.

            Why you have four strains tested in Figure 5., but in other measurement you have just AuNPs synthesized by two of them R. erythropolis and R. ruber? This two are also only one mentioned in Abstract. It could be due to better efficiency in biosynthesis. That should be explained through paper.

The other thing was about methods of characterization. Through the paper it is little bit confusing to understand which technique you use to characterize the bacteria and which to charactirze tha AuNPs. For example, look at second sentence in Abstract: “The effect of elevated (0.8– 15 3.2 mM) concentrations of НAuСl4 on viability, morphology and intracellular accumulation of AuNPs by different Rhodococcus species was shown.” It means that you measure viability of AuNPs and in the paper is clear that you tested on the beginning the viability of actinobacteria in presence of higher НAuСl4 concentration not the viability of AuNPs, what makes more sense because AuNPs are not alive.

This was the major things that I noticed other are listed in attachment.

Reviewer 2 Report

In this manuscript, the authors described the biosynthesis of a series of charged AuNPs via the enzymatic reduction induced by Rhodococcus. These bacteria strains were found to exhibit high resistance in the presence of intracellular cationic Au+3 in elevated concentrations. The AuNPs synthesized by these bacteria strains showed certain antimicrobial activities against M. luteus and E. coli (Figure 10). These results may be of interesting for publication after the revisions by taking some arguments and suggestions into consideration, based on the explanation of experimental results and conclusions.

1)      Authors evaluated chloroauric acid resistance of four Rhodococcus strains by comparing their cell viability in terms of the optical density of biomedia at 630 nm in Figure 1. Accordingly, there are several questions needed to be addressed as follows: (a) what type of compound optical absorbance are you following as the reference for comparison? If not the typical NADH reductases indicator via formazan formation for viability with the absorption at 570 nm (λmax of formazan); (b) most of the AuNPs themselves show absorption at 500–550 nm with the tail band extending to 700 nm. Will intracellular AuNPs formed (in both dead cells and viable cells) lead to misleading reading of OD630?; (c) other factors including light scattering by nanoparticles should be taken into account to deviate OD630; (d) additional experiments such as flow cytometry may be appropriate to support or strengthen the conclusion.

2)      With similar reasons to (1), antimicrobial activities of AuNPs on both Gram-positive and Gram-negative bacteria (Fig. 10) should apply alternative cell viability methods for accuracy, even though the cytotoxicity in the figure were in the low range for both cases.

3)      Notably in Fig. 7 showing slight positive zeta charge potential from the neutral value for AuNPs synthesized by R. erythropolis vs. the negative one by R. ruber. However, Table 1 indicated only positive counter ions of Na+ (high), Ca+2 (high), Mg+2, etc. in a nearly identical value range with no likely negative counter ions to reflect the quantity of corresponding surface charges of AuNPs. These data do not consistent with Fig. 7.

4)      Some sentences need to be improved for better understanding, including typos to be revised, such as sentences in line 119 to 123 not consistent with the figure caption of Fig. 5.

Reviewer 3 Report

Greetings, Editor: Thank you for providing me with the opportunity to review the article. I reviewed the article with the title: Biosynthesis and Characterization of Gold Nanoparticles Produced Using Rhodococcus Actinobacteria at Elevated Chloroauric Acid Concentrations. Overall, the article structure and content are suitable for the IJMS (ISSN 1422-0067) journal. I am pleased to send you major level comments. There are some serious flaws which need to be corrected before publication. Please consider these suggestions as listed below.

  1. The title seems OK.
  2. The abstract seems to be good. Please add one more introductory line to your objective at the beginning of the abstract.
  3. Research gaps should be filled in a more clear way with directed necessity for future research work.
  4. The introduction section must be written in a higher quality way, i.e., with more up-to-date references addressed.
  5. In the introduction, Page 1, at the end of Line 40, needs a reference. Please cite this article: Role of nanomaterials in the treatment of wastewater: A review.
  6. The novelty of the work must be clearly addressed and discussed. Compare previous research with existing research findings and highlight novelty.
  7. What is the main challenge?
  8. In the introduction, Page 1 Line 43 needs another reference. Please cite this article here with the existing reference 1-Recent advances in metal-decorated nanomaterials and their various biological applications: a review
  9. Please check the abbreviations of words throughout the article. All should be consistent.
  10. What is the problem statement?
  11. The main objective of the work must be written in a more clear and more concise way at the end of the introduction section.
  12. Please include all chemical and instrumentation brand names and other important specifications.
  13. Please provide a space between the number and units. Please revise your paper accordingly since some issues occur in several spots in the paper.
  14. The overall result section is well explained, but the arrangement is very weird.
  15. Regarding the replications, the authors confirmed that replications of experiments were carried out. However, these results are not shown in the manuscript. How many replicates were carried out by experiment? The results seem to be related to a unique experiment. Please clarify whether the results of this document are from a single experiment or an average resulting from replications. If replicated were carried out, the use of average data is required as well as the standard deviation in the results and figures shown throughout the manuscript. In the case of showing only one replicate, explain why only one is shown and include the standard deviations.
  16. Please add a comparative discussion section. It would be better for the reader.
  17.  
  18. Section 5 should include a conclusion and future perspectives.The conclusion section is missing some perspectives related to the future research work, quantifies the main research findings, and highlights the relevance of the work with respect to the field aspect.
  19. To avoid grammar and linguistic mistakes, major level English language proficiency should be thoroughly checked. Please revise your paper accordingly since several language issues occur at several spots in the paper.
  20. Reference formatting needs careful revision. All must be consistent in one format. Please follow the journal guidelines.
  21. Please follow the MDPI guide lines to prepare your articles. It was very difficult to read and understand. I also added some concluding remarks.

Round 2

Reviewer 2 Report

The response and revision are good, please see attachment for details.

Reviewer 3 Report

Accepted in the present form.